# A Process Evaluation of a Multi-Component Intervention in Dutch Dietetic Treatment to Improve Portion Control Behavior and Decrease Body Mass Index in Overweight and Obese Patients

**DOI:** 10.3390/nu10111717

**Published:** 2018-11-09

**Authors:** Willemieke Kroeze, Frédérique Rongen, Michelle Eykelenboom, Wieke Heideman, Claudia Bolleurs, Ellen Govers, Ingrid Steenhuis

**Affiliations:** 1Department of Health Sciences, Faculty of Sciences, Vrije Universiteit Amsterdam, Amsterdam and Amsterdam Public Health Research Institute, 1081 HV Amsterdam, The Netherlands; f.c.rongen@vu.nl (F.R.); m.eykelenboom@vu.nl (M.E.); wiekeheideman@hotmail.com (W.H.); 2Department Care for Nutrition and Health, School of Nursing, Christian University of Applied Sciences, 6717 JS Ede, The Netherlands; 3Dutch Association of Dietitians, De Molen 93, 3995 AW Houten, The Netherlands; claudiabolleurs@hotmail.com; 4Dutch Knowledge Centre of Dietitians on Obesity (KDOO), 1065 AC Amsterdam, The Netherlands; egovers@amstelring.nl

**Keywords:** multicomponent intervention, dietitians, process evaluation, portion control strategies, weight management, portion size

## Abstract

The SMARTsize intervention embeds an evidence-based portion control intervention in regular dietetic care. This intervention was evaluated to explore (1) which patients participated, (2) the implementation process, and (3) the outcomes of the intervention. The intervention was evaluated with an observational study design including measures at baseline, and three, six, and nine months after the start of the program. Data concerning the process (participation, dose delivered, dose received, satisfaction) and the outcomes (self-efficacy, intention, portion control strategies, and Body Mass Index (BMI) were collected with forms and questionnaires filled out by dietitians and patients. Descriptive analyses, comparison analyses, and cluster analyses were performed. Patients were mainly obese, moderately to highly educated women of Dutch ethnicity. Use of the intervention components varied from 50% to 100% and satisfaction with the SMARTsize intervention was sufficient to good (grades 7.2–8.0). Statistically significant (*p* < 0.001) improvements were observed for self-efficacy (+0.5), portion control strategies (+0.7), and BMI (−2.2 kg/m^2^), with no significant differences between patients with or without counselling. Three clusters of patients with different levels of success were identified. To conclude, implementing an evidence-based portion control intervention in real-life dietetic practice is feasible and likely to result in weight loss.

## 1. Introduction

There is overwhelming evidence that overweight and obesity have a negative influence on public health. Consequently, many weight loss programs have been developed and evaluated in clinically controlled settings, often resulting in effective weight loss programs, at least in the short-term [1]. However, the implementation of such evidence-based programs in real-life practice is challenging [2]. This could be due to the so-called gap between research and practice. Therefore, the Dutch government stimulates cooperation between research and practice via practice-based research [3]. This implementation study is a result of a collaboration between research and dietetic practice.

In 2014, the effects of the PortionControl@Home intervention were published [4,5]. This intervention has a unique focus on portion control strategies for weight loss with an emphasis on portion size and calorie density of portions [6,7]. The self-management intervention is not about dieting, but about sustainable changes in dietary behavior and incorporated several behavior change strategies [4]. Participants of the randomized controlled trial liked the program and the intervention resulted in significantly more weight loss compared to the control group. However, weight loss was not maintained at follow-up after six or 12 months [5]. It has been suggested that face-to-face contact has added value for the effectiveness of interventions and the maintenance of behavior change [8] as well as continued professional care in the prevention of weight regain [9,10].

Individual counseling provided by a dietitian is an effective treatment to lower Body Mass Index (BMI), with greater reductions being observed in those treated for at least six months [11]. However, basic health care insurance for dietetic treatment in the case of overweight or obesity without comorbidities is limited to 3 h a year in the Netherlands, while financial restraints can prevent patients from engaging in additional consultations and prolonged treatment [12].

Furthermore, the need to incorporate behavior change techniques in dietetic treatment has been expressed [13,14]. In practice, available guidelines for dietetic treatment strongly emphasize the nutritional part of dietetic treatment (e.g., the provision of nutritional information on what and how much nutrients or products one should eat or should not eat tailored to the patients’ needs), while guidelines for behavior change counselling are limited (e.g., strategies such as self-monitoring, goal-setting, portion control strategies, and relapse prevention that are required to manage consumption as well as the determinants and sub-behaviors that precede consumption).

Combining the evidence-based self-management intervention PortionControl@Home, which includes behavior change techniques, with an extended period of individual face-to-face counseling from a dietitian could be a feasible solution for maintaining adequate portion control strategies and weight loss. Therefore, we implemented this multi-component intervention, referred to here as the SMARTsize intervention, in cooperation with dietitians.

Since implementing an evidence-based intervention into regular dietetic care is a challenge, a thorough process evaluation is required to inform the development of future implementation tools. The aims of this process evaluation are to explore (1) which patients in the setting of a dietary practice participated in the SMARTsize intervention, (2) the implementation process of the SMARTsize intervention by dietitians (dose delivered, dose received, and satisfaction), and (3) a first impression of the effectiveness of the SMARTsize intervention in dietary practices. Knowledge derived from this study can be used to further develop dissemination and implementation strategies of this intervention.

## 2. Materials and Methods

### 2.1. Design and Procedure

This intervention study had an observational design with repeated measures at baseline (T0) and three, six, and nine months after the start of the program (T1, T2, and T3, respectively). After the baseline measurement, and 0.5 h of individual intake, patients participated in phase I (a 12-week period) of the intervention, which involved the website, home-screener, self-management workbook, and cooking classes. This period was followed by phase II, a period of six months with 2.5 h of additional individual consultations. Data concerning the process evaluation were collected with registration forms filled out by the dietitians during program delivery (dose delivered) and with questionnaires filled out by patients right after phase I (T1, dose received and satisfaction) and at T2 (satisfaction additional individual consultations). To evaluate the outcomes of the intervention, self-reported data of patients on self-efficacy and intention (T0, T1, and T2), portion control strategies (T0 and T1), and BMI (T0, T1, T2, and T3) were collected. Dietitians received clear instructions on data registration. The evaluation was based on the RE-AIM model [15] and the approaches of Saunders [16] and Grol and Wensink [17]. The medical ethical committee of VU University Medical Center Amsterdam declared that the Medical Research Involving Human Subjects Act does not apply to the study and waived the need for approval (letter dated 25 May 2015; registration No. 2015.194).

### 2.2. Recruitment of Participants

The recruitment of participants was conducted in two stages. At first, dietitians who were interested in working with SMARTsize were recruited through an online newsletter that was distributed among members (approximately 2800 out of 3044 certified dietitians in The Netherlands) of the Dutch Association of Dietitians. Furthermore, open recruitment strategies using LinkedIn and Facebook were used to reach Dutch dietitians without a membership. The newsletter provided a hyperlink to an online survey about determinants of evidence-based dietetic practices. A gift card of 5 euros, which could also be donated to charity, was offered as incentive to the first 175 dietitians that completed the questionnaire. In the last part of the questionnaire, the evidence-based SMARTsize intervention was introduced and the possibility to implement this in practice was offered. Dietitians were asked whether they were willing to participate in a study about the implementation of SMARTsize. Free accredited training to work with SMARTsize was offered as an incentive. Of the 322 respondents of the online survey, 180 dietitians applied for more information about the implementation study. In addition, 48 dietitians contacted the SMARTsize team directly to ask for more information. The eligibility of dietitians was assessed through telephone calls with the principal researcher (WH) with all dietitians, or a delegate of a group of dietitians. To check the feasibility of delivering the SMARTsize intervention, dietitians had to agree to try to recruit seven patients and deliver three cooking classes. Finally, 62 dietitians expressed an interest in the study. However, due to budget restraints we included 43 dietitians covering a variety of urban and more rural regions in the Netherlands. They all signed informed consent and declared the intention to deliver the SMARTsize intervention to approximately seven overweight patients. Dietitians were not financially compensated by the project team for the delivery of the cooking classes, but had to find their own way to finance this intervention component. If needed, dietitians received reimbursement for rental fees of the cooking class venue, up to a maximum of 150 euros.

Dietitians recruited their own patients for the SMARTsize intervention who were also willing to participate in the research project. Patients were provided with clear information on the intervention and their participation in the study following the ethical standards for research. Eligibility criteria for patients were: patients currently not receiving treatment for weight loss with a BMI ≥25. Exclusion criteria were: eating disorder, renal failure, heart failure, insulin uses >2 dosages per day. As an incentive, patients received a free copy of the ‘SMARTsize book’ [18] which was one of the intervention components. The individual consultations were financially reimbursed by the patients’ health care insurer.

### 2.3. Intervention Materials

Several dissemination and implementation strategies were applied to deliver the intervention. All dietitians received free, accredited training in the principles of the SMARTsize intervention through an e-learning course (approximately 15–20 h study load) and a one-day group training on skills and strategies for relapse prevention in individual counseling. Both the e-learning and training were delivered by I.S. and her associate. Furthermore, dietitians were provided with a detailed manual with background information and dissemination and implementation strategies for the SMARTsize intervention together with all intervention materials. Dissemination strategies included information leaflets for potential patients, an example of a press release, and ideas on how to recruit new patients in addition to their standard routines. Implementation strategies included clear instructions on how to organize and conduct the cooking lessons from finding a suitable venue, to arranging groceries, to recipes, to educational activities and strategies during the cooking lessons. Furthermore, dietitians received instructions on how to incorporate relapse prevention in their individual counselling. These instructions were based on theory and evidence derived from Larimer and Marlatt [19,20]. Additionally, a help-desk from the research team (reachable by email and telephone) was available for questions from both dietitians and patients concerning the use of the online tool or the organization of the cooking classes. In addition, a closed Facebook group was created for the dietitians to exchange knowledge and ideas and social support. Finally, five newsletters containing information about implementation and study progress and procedures were sent by the research team to keep the dietitians involved and motivated.

The SMARTsize intervention aims to stimulate and improve adequate portion control behavior in order to decrease energy intake and body weight. The SMARTsize intervention is a multicomponent intervention based on the evidence-based 12-week program PortionControl@Home [4] and 3 h of individual face-to-face counseling with a dietitian. The treatment started with an individual 30-min intake to assess eligibility and explain the procedure and goals of the treatment. The intake was followed by phase I (a 12-week period) of the intervention. Intervention components were offered gradually, starting with the website, including an online tool to increase knowledge and awareness of portion sizes. This tool was a restyled and improved version of the PortionSize@awarenessTool. In week 2, patients received a self-management workbook about portion control strategies based on behavior change techniques (e.g., monitoring, goal-setting, action planning, coping planning). The cooking classes, guided by the dietitian, were offered during weeks 3–10 and patients received the portion control home-screener (online via the secured website or optional in paper-and-pen format) after the last cooking class. A clear and elaborate description of the intervention components and information on the development and evaluation can be found elsewhere [4,21,22]. After phase I, 2.5 h of additional individual face-to-face consultations with the dietitian were scheduled in the course of six months. Dietitians were allowed to plan and conduct their individual consultations following their professional expertise and usual practice. In addition, dietitians were encouraged to pay special attention to relapse prevention and maintaining adequate portion control behavior.

### 2.4. Data Collection and Measures

Sociodemographic and professional characteristics of dietitians were collected with an online questionnaire after the training. Data included: sex, age, years of working experience, estimated hours of patient care per week. Sociodemographic and health characteristics of patients were collected with a paper-and-pen questionnaire at T0. Data included: age, sex, nationality, marital status, employment, educational level, health status (if patients suffered from diabetes mellitus type 2, cardiovascular disease, high blood pressure, and/or high cholesterol in the past 12 months), weight status, weight loss history and previous dietetic counseling (did you ever have the guidance of a dietitian before?).

The implementation process of the SMARTsize intervention was monitored by data on the actual offer of the various intervention components (dose delivered), the use of these intervention components (dose received) and patients’ satisfaction with the components. Delivery of the online tool was recorded by dietitians on the consultation form of each individual patient (personal account created: yes/no). Delivery of the workbooks was recorded by the research team after sending the book by mail to the patient. Activation of the online home-screener was derived from logs of the website. The number of cooking lessons offered to each patient was registered by the dietitian. Date and direct treatment time (duration) of every individual consultation were registered by dietitians in consultation reports as well as topics related to relapse prevention discussed during individual counseling (weight maintenance, identify difficult situations, what to do in difficult situations, and what to do with relapse). Reasons for drop-out of patients were registered by the dietitian. Use of the different components of SMARTsize was measured at T1 with a self-reported questionnaire for patients, asking if: the website was used (yes/no), the book was read (completely, partially, not), the home-screener was used (yes/no), and the number of cooking classes attended (1, 2, 3, or none). The attendance rates at cooking lessons were also registered by the dietitian. From these numbers the percentage of actual use of the offered intervention components was calculated. General satisfaction with each of the different SMARTsize intervention components were graded by patients on a scale from 1–10 (only round grades could be given). In addition, the usefulness, reliability, innovativeness, and understandability of each component were evaluated by patients using a five-point Likert scale (ranging from 1 = totally disagree to 5 = totally agree).

Outcome evaluation measurements were self-efficacy, intention, portion control strategies, and BMI. Self-efficacy towards paying attention to portion sizes of food and beverages and self-efficacy towards preparing usual dishes with less calories were measured on a five-point scale (ranging from 1 = definitely not able to 5 = definitely able). In addition, intention to regularly consume smaller portions of food and beverages was measured on a five-point scale (ranging from 1 = no intention at all to 5 = intention). The use of portion control strategies was measured with a validated instrument [21] containing 32 items on different portion control strategies measured on a five-point scale (ranging from 1 = (almost) never using strategy to 5 = (almost) always using strategy). From these items, the average use of portion control strategies was calculated. Dietitians measured and registered height and weight at T0 and weight during every individual consultation. Furthermore, patients reported their weight in the questionnaires of T1, T2, and T3. Since there were many missing weight records in the registration of dietitians and there was no significant difference between registered and self-reported weight, we calculated BMI based on self-reported data from patients at all time-points.

### 2.5. Statistical Analyses

Baseline characteristics of dietitians and patients were explored with descriptive analyses. Furthermore, baseline differences between patients who dropped-out from the intervention before additional individual counseling and patients who did receive the individual counseling were explored with ANOVA analyses.

For the process evaluation, data on dose delivered (percentage of respondents to whom SMARTsize components were offered and various topics on relapse prevention were discussed as well as average treatment time (mean, standard deviation (SD)) of individual consultation), dose received (percentage of the actual use of offered components), and satisfaction (mean, SD) with the various intervention components were descriptively analyzed. In addition, dose delivered was qualitatively compared to protocol. Furthermore, differences in satisfaction with the different SMARTsize components between patients with and without additional individual counseling were explored with independent *t*-tests.

The outcomes of the SMARTsize program (self-efficacy of smaller portions, self-efficacy of low-calorie dishes, intention to have smaller portions, portion control strategies, and BMI) were described (mean, SD) and tested with paired-samples *t*-tests. ANOVA analyses were used for differences between patients with and without additional individual counseling.

Furthermore, patterns of weight change during phase I (T0–T1) and phase II (T1–T3) were identified by conducting a two-step cluster analysis with a log-likelihood distance measure. Before determining the cluster solution, the assumptions to perform a log-likelihood distance measure were checked. The normality of percent weight change in each phase was investigated using plots and the independence of both weight change variables was tested using the Pearson correlation coefficient. The optimal number of clusters was determined using the Bayesian Information Criterion (BIC) [23]. The goodness of the final cluster solution was tested using the silhouette coefficient for cohesion and separation [23]. Silhouette coefficients ≥0.5 indicate a good cluster quality [23]. To examine the stability of the final cluster solution, the cluster analysis was conducted a subsequent four times with the cases arranged in four random orders. In addition, the study sample was divided into random halves, followed by repeated cluster analysis on each half [24]. Cohen’s kappa coefficients were used to assess the agreement between the original cluster solution and the clusters formed by the two methods to check for stability [25,26]. Descriptive statistics were used to characterize the identified clusters. Differences between the weight change patterns were tested using chi-square analyses and univariable multinomial logistic regression analyses.

## 3. Results

### 3.1. Description of Participants

The characteristics of dietitians (*n* = 43) who were included in the study are presented in Table 1. The mean age was 41.8 ± 9.9 years and the median category of working experience years was 11–15 years. All dietitians received training about relapse prevention, and 39 dietitians (90.7%) completed the e-learning SMARTsize certification. During the enrollment period of patients (between August 2015 and April 2016), eight dietitians dropped out because they were not able to recruit patients. Reasons for drop-out were: maternity leave (*n* = 1), working in a neighborhood with many people with a migrant background or a low socioeconomic status (*n* = 4), patients preferring regular treatment or not able to find patients who met the inclusion criteria (*n* = 3). During the intervention period, another six dietitians dropped out due to low patient numbers (*n* = 4), patients not interested in individual counselling (*n* = 1), or no possibility to provide individual counseling due to setting (*n* = 1). This resulted in 29 dietitians who offered the complete SMARTsize intervention.

Dietitians enrolled 228 overweight patients, with an average of 6.5 patients per dietitian. Three patients withdrew before the start due to personal circumstances, leaving 225 patients at baseline (Table 2). The mean age for all patients was 49.7 ± 12.7 years, and the majority was female (76.0%), moderately (40.6%) or highly (38.8%) educated, with a mean BMI of 33.0 ± 5.4 kg/m^2^.

During the first 12 weeks, 66 patients (29%) dropped out for various reasons including: personal/medical reasons (*n* = 10), because expectations were not satisfied (*n* = 3), due to non-specified issues with the SMARTsize components (*n* = 1), no motivation to continue counseling (*n* = 4), starting another program (*n* = 2), did not want to start counseling for financial issues (*n* = 1), or unknown reasons (*n* = 45). Participants who dropped out before individual consultations did not significantly differ in baseline characteristics from participants who did receive individual consultations.

### 3.2. Process Evaluation of the Implementation Process

Results of dose delivered and dose received are presented in Table 3. All patients received access to the website and the workbook. Most patients (89%) indicated that they used the website, 39% indicated they read the workbook completely and 43% read the workbook partially. The home-screener was activated for the majority of the patients (70%), but distribution of the optional ‘hardcopy home-screener’ was not recorded. Less than half of the respondents (45%) indicated they used the home-screener. Out of the 122 patients that were offered three cooking lessons, 45.9% attended all lessons. Additional individual counselling was received by 156 patients, with a total average duration of 78.6 ± 45.3 min. On average, the total duration of the entire SMARTsize intervention was 25.9 ± 8.11 weeks.

In general, patients were satisfied with the SMARTsize intervention (Table 4). Grades for the SMARTsize components varied between 7.2 and 8.0 on a 10-point scale. The intervention components were perceived as useful, reliable, moderately innovative, and understandable. Patients who received all intervention components including additional individual counseling rated the usefulness of the information on the website (4.2 versus 3.8), the overall grade for the workbook (8 versus 7.5), and innovativeness of the information in the book (3.6 versus 3.2) significantly higher compared to participants who did not receive individual counselling.

### 3.3. Outcome Evaluation of the SMARTsize Intervention

Scores on self-efficacy to prepare low calorie dishes were statistically significantly higher at T2 (3.9 ± 0.8) compared to T0 (3.4 ± 1.0) (*p* < 0.001). Intention to eat smaller portions was statistically significantly lower at T2 (4.3 ± 0.9) compared to T0 (4.5 ± 0.6) (*p* < 0.001). The use of portion control strategies was statistically significantly higher at T1 (3.7 ± 0.5) compared to T0 (3.0 ± 0.5) (*p* < 0.001). Self-reported BMI statistically significantly decreased from 33.0 ± 5.4 at T0 to 30.8 ± 4.3 at T3 (*p* < 0.001). All scores did not significantly differ between patients with or without additional individual counselling (Table 5).

A total of 93 patients were included in the cluster analysis, after excluding all patients with a missing value at T0, T1, or T3. Missing data analysis using chi-square and independent-sample *t*-tests indicated that sex, age, educational level, BMI at baseline, and attended cooking classes did not significantly differ between participants who did or did not report their weight at T0, T1, and T3 (*p* > 0.05). However, missing data analysis showed that participants who had a missing value of weight attended less consultations and were less satisfied than those who reported their weight at all time-points. The two-step cluster analysis identified three clusters of participants based on percent weight change during phase I and phase II. Results are shown in Figure 1. The largest cluster was labeled as “moderately successful”. The cluster comprised 64.5% of the participants (*n* = 60) and was associated with a mean percent weight change of −1.67 (SD = 1.96) during phase I and −2.15 (SD = 3.02) during Phase II (Figure 1). The second largest cluster comprised 19.4% of the participants (*n* = 18) and was associated with a mean percent weight change of −7.13 (SD = 1.88) during phase I and −5.40 (SD = 2.50) during phase II (Figure 1). This cluster showed initial weight loss followed by continued weight loss and was therefore labeled as “highly successful”. Lastly, the smallest cluster comprised 16.1% of the participants (*n* = 15) and was associated with a mean percent weight change of −3.55 (SD = 3.51) during phase I and +5.04 (SD = 2.37) during phase II (Figure 1). This cluster showed weight loss followed by complete regain and was therefore labeled as “less successful”. The silhouette measure for cohesion and separation showed good cluster quality. Subsequent cluster analyses with the cases arranged in four random orders demonstrated almost perfect stability (kappa = 1.00), and cluster analyses on random halves of the sample demonstrated the substantial stability of the clusters (kappa = 0.61, kappa = 0.78). Compared to the moderately successful cluster, the highly successful cluster was negatively associated with age (Odds Ratio (OR) = 0.93, 95% Confidence Interval (CI) = 0.880–0.98) and positively associated with BMI at baseline (OR = 1.18, 95% CI = 1.05–1.33). There were no other statistically significant differences in patient characteristics between the three weight loss clusters (Table 6).

## 4. Discussion

The aim of our study was to explore the implementation process of an evidence-based portion control intervention in regular dietetic care. The main findings with respect to patients reached, the implementation process, and first impressions of the effectiveness are discussed below.

Patients participating in the SMARTsize intervention offered by dietitians were mainly obese, moderately to highly educated women. Our patient population was more or less comparable to the participants of the randomized controlled trial evaluating the effect of PortionControl@Home [5], although the current study included more men (24% versus 15%) and patients with a higher BMI (33.0 versus 32.4) and the average age was 4 years older (49.7 versus 45.7 years). In comparison, the general patient population of dietitians in 2016 had an average age of 53 years and an average BMI of 32.9 and included 36% males and 24% lower educated patients [27]. The relatively low number of men in dietetic interventions could be explained by research findings that men are less likely to feel a need for help with weight loss compared to women [28,29]. With respect to education level, dietitians indicated that they considered the intervention to be less suitable for patients with low levels of health literacy and therefore did not recruit such patients [30].

With respect to the implementation process, all patients used the website and the book, almost half of the patients used the home-screener, and over half of the patients attended at least two cooking classes. Furthermore, 70% of the patients received individual counselling. Perceived satisfaction with the SMARTsize intervention was sufficient to good, with no differences between patients who dropped out before individual counseling and patients who did receive counseling. Half of the dietitians did not offer three cooking classes, but instead offered a series of two lessons. Dietitians indicated that a main barrier for implementation of the cooking classes was the time-consuming organization of the cooking classes without financial reimbursement [30]. Despite this practical barrier, dietitians did acknowledge the added value of the cooking classes [30]. In addition, the home-screener was not activated by the dietitians for 30% of the patients. Dose received was comparable with findings from the evaluation of PortionControl@Home [5]: 89% used the website (versus 90.4%), 45.9% attended three cooking classes if offered (versus 40.3%), and 44.6% used the home-screener (versus 48.2%). The exception was the use of the book; only 38.7% read the book completely compared to 64% in the previous study. In addition, only 70% received additional individual consultation with a total treatment time of 79 min, while 150 min were proposed.

Although this study was not designed as an effect study, first indications of effectiveness were explored. The self-efficacy of preparing dishes with less calories was significantly higher at follow-up. This is an important result, since a recent systematic review on determinants of weight loss maintenance reported that self-efficacy is one of the determinants with the strongest level of evidence [31]. A good self-efficacy towards diet, physical activity, or weight management will result in a better energy-balance and weight maintenance after initial weight loss [31,32]. In addition, the use of portion control strategies was higher at follow-up. This improvement was even higher, with 0.7 compared to 0.49 in the previous study [5], which is a promising result, since there is strong evidence that portion control is an important determinant of weight maintenance [31].

The body mass index of patients in the current study decreased over time from 33.0 at baseline to 31.9 at three months to 30.9 at nine months. These results are better than results of regular dietetic care, where BMI decreases by on average 0.94 points and by 1.83 points in patients treated for at least six months [11]. In contrast, Poelman and colleagues [5] reported a reduction in BMI after three months, followed by an increase after six and 12 months. This difference in effect between studies could be caused by the additional individual counseling offered in phase II of the current study. However, within the current study we did not find significant differences between patients who dropped out before individual counseling and patients who did receive additional counseling. This finding has also been reported in another study investigating the added value of individual counselling [33]. This lack of significance could be due to low numbers of patients without counseling and the low numbers of patients with complete data, resulting in insufficient power to detect differences. Another explanation could be related to the quantity and the quality of the individual counseling [34].

Unfortunately, we were not able to relate the implementation of the intervention to the effectiveness of the intervention due to an excess of missing values. Therefore, we cannot draw any conclusions on a dose–response relationship or the added value of additional individual counseling. However, a review [9] on the impact of extended care (defined as at least two sessions delivered by a trained interventionist, focusing on continuing support for behaviors associated with weight management) concluded that extended care would lead to the maintenance of an additional 3.2 kg weight loss over 17.6 months post intervention compared with a control group. It must be noted that the majority of included extended care interventions in this review lasted for at least 12 months, which is substantially longer than the six months in the current study.

Furthermore, there were three weight loss clusters defined (low, moderate, and high success), with patients in the high success cluster being significantly younger and having a higher BMI at baseline compared to the moderate success cluster. A higher initial BMI has previously been associated with greater weight loss [11]. This could be caused by the fact that patients with a higher BMI have a higher metabolic rate, resulting in more weight loss when energy intake is reduced. In addition, patients with a higher BMI might feel a stronger sense of urgency to lose weight [28]. It could also be the case that younger patients were more capable of using the digital intervention components. Furthermore, younger patients might have been more physically active or engaged in more sports compared to older patients [35]. Unfortunately, we did not measure this. Our findings are largely supported by a systematic review reporting consistent evidence that age, gender, socioeconomic status, and weight history are not significant in predicting weight loss maintenance [31].

This study has several limitations that need to be considered. A main limitation is the absence of a control group. Therefore, we cannot conclude that the SMARTsize intervention is more effective than care as usual. Furthermore, participants volunteered to participate in this study and were not randomly selected. However, our participants were comparable with the general patient population of dietitians in primary care. Another limitation is the amount of missing data, resulting in a lack of power to test for significant differences. This prevented us from conducting meaningful dose–response analyses to explore the added value of the different intervention components. Finally, self-reported BMI was used to evaluate the outcomes of the intervention. Self-reported BMI is associated with under-reporting by 0.36 kg/m^2^ [36]. However, self-reported BMI did not differ significantly from BMI measured by dietitians and it appeared to be a valid alternative in the previous study [5]. Lastly, social desirability in the answers of patients can also be a risk. This risk might have been reduced by the fact that dietitians implemented the intervention and the questionnaires were handled by the researchers. Patients were made aware of the fact that dietitians did not receive their answers.

To conclude, it is feasible to implement the evidence-based portion control SMARTsize intervention into regular dietetic care. Weight loss as a result of this intervention, plus the added value of individual counseling, is likely. However, the reach of the SMARTsize intervention could be improved with respect to lower educated patients and males, while there is also room for improvement in the offer and use of the home-screener, the cooking classes, and the individual consultations.

## Figures and Tables

**Figure 1 nutrients-10-01717-f001:**
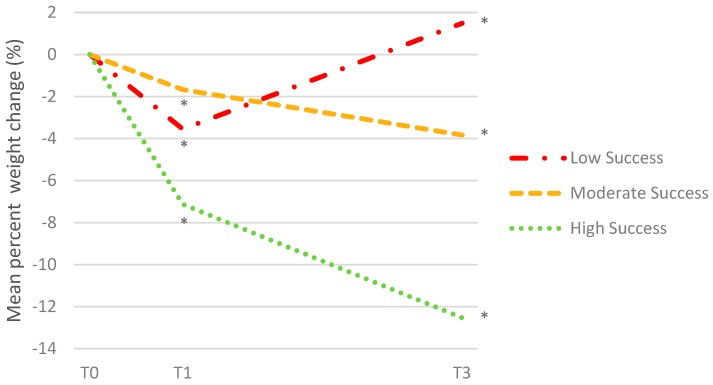
Patterns of weight change over nine months identified by cluster analysis. T0 = baseline measurement; T1 = three-month follow-up; T3 = nine-month follow-up. * *p* < 0.001.

**Table 1 nutrients-10-01717-t001:** Personal and professional characteristics of dietitians (*n* = 43).

Gender (female)	97.7%
Age (mean ± SD years)	41.8 ± 9.9
Experience	
0–5 years	25.6%
6–10 years	10.3%
11–15 years	33.3%
16–20 years	7.7%
21–25 years	10.3%
26–30 years	5.1%
>32 years	7.7%
Patient care (mean ± SD hours per week)	20.7 ± 7.9
Training	
E-learning SMARTsize completed	90.7%
Training on relapse prevention	100%
Patients included per dietitian (mean (range))	6.5 (1–17)

**Table 2 nutrients-10-01717-t002:** Sociodemographic and weight-related characteristics of patients (*n* = 225) at baseline.

		Participants in SMARTsize Intervention		
		Total (*n* = 225) ^a^	No Additional Counseling (*n* = 66) ^b^	Additional Counseling (*n* = 159) ^c^
Age	Mean ± SD	49.7 ± 12.7	51.5 ± 13.0	48.9 ± 12.6
Men	%	24.0	27.3	22.6
Women	%	76.0	72.7	77.4
Dutch ethnicity	%	94.9	96.6	94.2
Non-western ethnicity	%	5.1	3.4	5.8
Married or with partner	%	79.0	76.3	80
Household number of people	Mean ± SD	2.9 ± 0.8	2.7 ± 1.2	3.0 ± 1.3
Employed (yes)	%	93.9	94.9	93.5
Education level: low	%	20.6	23.7	19.4
Education level: middle	%	40.7	30.5	44.5
Education level: high	%	38.8	45.8	36.1
Diabetes mellitus type 2 (yes)	%	9.7	15.1	7.9
Cardiovascular disease (yes)	%	8.3	16.4	5.3
High blood pressure (yes)	%	20.1	28.8	15.7
High cholesterol (yes)	%	19.7	21.1	17.6
Weight (self-reported)	Mean ± SD	97.4 ± 17.6	98.2 ± 18.6	97.2 ± 16.9
BMI ^d^ (kg/m^2^) (self-reported)	Mean ± SD	33.0 ± 5.4	32.9 ± 4.5	33.2 ± 5.1
BMI 25 ≤ 30 kg/m^2^	%	30.2	28.9	30.6
BMI 30 ≤ 40 kg/m^2^	%	60.9	62.2	60.5
BMI > 40 kg/m^2^	%	8.9	8.9	8.9
Previous attempts losing weight (yes)	%	85.0	93.2	81.9
Perceived weight development in past five years				
Same weight	%	6.5	11.1	4.5
Gained weight	%	48.6	52.4	47.1
Decreased weight	%	4.2	4.8	4.5
Fluctuating weight	%	40.7	31.7	43.9
Previous counselling of a dietitian (yes)	%	56.0	51.7	57.7

^a^ The range of valid cases varied between 198 and 225; ^b^ the range of valid cases varied between 41 and 66; note that patients with and without additional counseling did not significantly differ on all characteristics; ^c^ the range of valid cases varied between 146 and 159. ^d^ BMI is Body Mass Index.

**Table 3 nutrients-10-01717-t003:** Process evaluation (dose delivered and received) of treatment components.

Dose Delivered	*n*	%	Dose Received	*n*	% of Dose Delivered
SMARTsize Components			Use of Components		
Website (account created)	225	100	Website (yes)	171	89.1
Book	225	100	Read book completely	87	38.7
			Read book partially	96	42.7
Home-screener (activated)	155	70.0	Home-screener (yes)	86	44.6
Cooking classes offered ^a^			Attendance cooking classes ^b^		
2	65	34.8	0	6	9.2
			1	21	32.3
			2	38	58.5
3	122	65.2	0	11	9.0
			1	21	17.2
			2	34	27.9
			3	56	45.9
**Additional Counseling ^e^**					
**Number of Consultations ^d^**	***n***	**%**		***n***	**%**
1	29	18.6	No individual counseling	69	30.7
2	28	17.9			
3	44	28.2			
4	34	21.8			
5	12	7.7			
6	5	3.2			
7	4	2.6			
Treatment time (minutes) mean (SD) ^c^				78.6 (45.3)	
**Relapse topics discussed during counseling:**				***n***	**%**
Weight maintenance				113	72.0
Identify difficult situations				109	69.4
What to do in difficult situations				107	68.2
What to do with a relapse				91	58.0

^a^ The number of cooking classes delivered is the number of classes offered by dietitians; ^b^ the number of cooking classes attended as registered by dietitians. This is the attendance number of the actual offer; ^c^ treatment time counseling was registered for 124 patients; ^d^ 156 patients (69.3%) received counseling; ^e^ the number of consultations received is the same as the number of consultations offered.

**Table 4 nutrients-10-01717-t004:** Satisfaction of patients with components SMARTsize treatment.

	Participants in SMARTsize Intervention					
	Total		No Additional Counseling		Additional Counseling	
	*n* ^a^	Mean (SD)	*n* ^a^	Mean (SD)	*n* ^a^	Mean (SD)
**Website**						
Grade ^b^	171	7.2 (1.4)	41	6.9 (1.7)	130	7.3 (1.3)
Information useful ^c^	173	4.1 (0.9)	41	3.8 (0.9)	132	4.2 (0.8) *
Information reliable ^c^	173	4.3 (0.7)	41	4.2 (0.6)	132	4.3 (0.7)
Information innovative ^c^	173	3.4 (1.0)	41	3.2 (1.2)	132	3.4 (0.9)
Information understandable ^c^	173	4.4 (0.7)	41	4.3 (0.7)	132	4.3 (0.7)
**Home-screener**						
Grade	86	7.2 (1.4)	23	7.0 (1.7)	63	7.3 (1.3)
Information useful	87	4.1 (0.8)	23	3.9 (0.9)	64	4.1 (0.8)
Information reliable	87	4.1 (0.7)	23	4.1 (0.9)	64	4.1 (0.7)
Information innovative	81	3.5 (1.0)	21	3.3 (1.3)	50	3.6 (0.9)
Information understandable	87	4.3 (0.7)	23	4.3 (0.7)	64	4.2 (0.7)
**Book**						
Grade	181	7.9 (1.3)	41	7.5 (1.4)	140	8.0 (1.2) *
Information useful	185	4.3 (0.8)	41	4.2 (0.9)	144	4.4 (0.7)
Information reliable	185	4.4 (0.7)	41	4.4 (0.6)	144	4.3 (0.7)
Information innovative	185	3.5 (1.0)	41	3.2 (1.3)	144	3.6 (1.0) *
Information understandable	185	4.5 (0.6)	41	4.5 (0.7)	144	4.5 (0.6)
**Cooking classes**						
Grade	163	8.0 (1.6)	30	7.5 (1.7)	133	8.2 (1.6)
Information useful	166	4.3 (1.0)	30	4.0 (1.1)	136	4.3 (0.9)
Information reliable	166	4.4 (0.8)	30	4.2 (0.9)	136	4.4 (0.8)
Information innovative	166	3.8 (1.2)	30	3.4 (1.3)	136	3.9 (1.1)
Information understandable	166	4.6 (0.7)	30	4.5 (0.7)	136	4.6 (0.6)
**Additional counseling ^a^**						
Grade (1–10)	-	-	-	-	119	7.8 (1.2)
Consultations were useful	-	-	-	-	120	4.1 (1.0)
Consultations were reliable	-	-	-	-	119	4.1 (1.0)
consultations innovative	-	-	-	-	120	3.5 (0.9)
Consultation were understandable	-	-	-	-	120	4.2 (0.9)

^a^*n* is number of valid cases; ^b^ grade was measured on a scale from 1 to 10; ^c^ the characteristics of being useful, reliable, innovative, and understandable were measured on a scale from 1 to 5. * Significant differences between patients with and without additional counseling (*p* < 0.05).

**Table 5 nutrients-10-01717-t005:** Outcomes of the SMARTsize treatment.

	Participants in SMARTsize Intervention											
	Total				No Additional Counseling				Additional Counseling			
	T0	T1	T2	T3	T0	T1	T2	T3	T0	T1	T2	T3
**Self-efficacy of smaller portions ^a^**												
Mean (SD)	3.5 (0.8)	3.7 (0.8) *	3.7 (0.8)	-	3.5 (0.8)	3.8 (0.9) *	3.6 (0.8)	-	3.5 (0.7)	3.7 (0.8) *	3.7 (0.8)	-
*n*	214	195	145		59	46	20		155	149	125	
**Self-efficacy of low-calorie dishes ^a^**												
Mean (SD)	3.4 (1.0)	3.8 (0.9) *	3.9 (0.8) *	-	3.4 (1.1)	3.8 (0.9) *	3.8 (1.0) *	-	3.4 (1.0)	3.8 (0.8) *	3.9 (0.8) *	-
*n*	212	194	145		58	46	20		154	148	125	
**Intention to intake smaller portions ^b^**												
Mean (SD)	4.5 (0.6)	4.4 (0.8)	4.3 (0.9) *	-	4.6 (0.6)	4.4 (0.9)	4.0 (1.1) *	-	4.5 (0.6)	4.4 (0.8)	4.4 (0.9) *	-
*n*	215	195	145		59	46	20		156	149	125	
**Portion control strategies ^c^**												
Mean (SD)	3.0 (0.5)	3.7 (0.5) *	-	-	3.1 (0.5)	3.7 (0.5) *	-	-	3.0 (0.5)	3.7 (0.5) *	-	-
*n*	216	195			60	46			156	149		
**BMI**												
Mean (SD)	33.0 (5.4)	31.9 (4.7) *	31.4 (4.8) *	30.8 (4.3) *	32.6 (5.0)	32.0 (5.2) *	32.4 (5.1) *	31.6 (3.3) *	33.1 (5.5)	31.9 (4.6) *	31.2 (4.8) *	30.7 (4.4) *
*n*	212	189	142	94	57	43	21	10	155	146	121	84

^a^ Self-efficacy of paying attention to portion sizes of food and beverages and of preparing usual dishes with less calories were measured on a five-point scale from 1 (definitely not able) to 5 (definitely able). ^b^ Intention to regularly consume smaller portions of food and beverages was measured on a five-point scale from 1 (no intention at all) to 5 (intention). ^c^ The mean of 32 items measured on a five-point scale from 1 ((almost) never using portion control strategies) to 5 ((almost) always using strategies). * Statistically significant difference between follow-up and baseline measurement, tested with paired-samples *t*-test (*p* < 0.05). Note that participants with and without additional counselling did not significantly differ on all outcomes, tested with ANOVA analysis (*p* < 0.05).

**Table 6 nutrients-10-01717-t006:** Descriptive statistics of weight loss clusters.

	Overall *n* = 93	Low Success ^a^ *n* = 15	Moderate Success ^a^ *n* = 60	High Success ^a^ *n* = 18
Age (years), mean (SD)	51.2 (12.0)	47.9 (9.2)	54.0 (12.3) ^b^	44.7 (10.1) ^b^
BMI at baseline, mean (SD)	32.4 (4.7)	32.0 (4.9)	31.6 (4.5) ^c^	35.5 (4.2) ^c^
Gender, *n* (%) ^‡^				
Female	71 (76.3)	10 (66.7)	47 (78.3)	14 (77.8)
Male	22 (23.7)	5 (33.3)	13 (21.7)	4 (22.2)
Education, *n* (%) ^‡^				
Low	13 (14.0)	1 (6.7)	8 (13.3)	4 (22.2)
Middle	38 (40.9)	5 (33.3)	26 (43.3)	7 (38.9)
High	42 (45.2)	9 (60.0)	26 (43.3)	7 (38.9)
Previous attempts at losing weight, *n* (%) ^‡^				
Yes	77 (82.8)	14 (93.3)	48 (80.0)	15 (83.3)
No	16 (17.2)	1 (6.7)	12 (20.0)	3 (16.7)

^a^ Success clusters based on percent weight change during phase 1 (T0–T1) and phase 2 (T1–T3); ^b,c^ significant difference (*p* < 0.05) between moderate and high success clusters derived from univariable multinomial logistic regression analyses. ^‡^ Chi-square analyses revealed no statistically significant differences between the clusters.

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
