# Peer review of "A Process Evaluation of a Multi-Component Intervention in Dutch Dietetic Treatment to Improve Portion Control Behavior and Decrease Body Mass Index in Overweight and Obese Patients"

_nutrients, 2018, doi:10.3390/nu10111717_

Round 1

Reviewer 1 Report

This is an interesting report of an important step in translating research findings into clinical practice. Obesity/overweight is an important medical condition and effective interventions are difficult to identify.  The model of implementing research proven findings in a clinical practice network is unique and potentially important.

The impact of the findings here are somewhat muted however by the design of the study.  First, long term weight loss is probably defined as something beyond 9 months but this study had its final time point at 9 months.  Second, the key measure of success would presumably be BMI which the text states is "self-reported" which limits its value.  Third, the drop out rate was significant limiting interpretation of results from a population that was not random to begin with.  Fourth, there is no control group limiting interpretation of the significance of weight loss.  Most dietary interventions will be successful in study settings so the question of efficacy of this intervention requires a comparator group.

Author Response

Dear reviewer, 

Please find our response to your comments in the attached PDF document under the heading 'reviewer 1.'

Sincerely,

the authors

Reviewer 2 Report

This manuscript describes finding s from a dietary intervention study in Netherlands.

Specific comments:

- Abstract: Would be useful to have some numerical data from main analyses

- Introduction: It is too long. Should be condensed.

- Results: Please avoid duplication of data already presented in Tables

- Table 3: Would there be better terms than "dose delivered" and "dose received". "Dose" is associated with medications

- Table 5 and discussion: It is a major limitation that there was no control group. It is difficult to say whether the changes in BMI are due to SmartSize or just because the patients had intervention visits. Therefore, you need to be very careful when discussing the effectiveness. It's not very useful to compare with other studies (ref 11), because study populations are always different.

- Study limitations should be more openly and thoroughly discussed

Author Response

There reviewer,

Please, find our response to your comments in the attached PDF document under the heading 'reviewer 2'.

Sincerely,

the authors

Round 2

Reviewer 1 Report

This revision directly acknowledges limitations identified in the initial review and adjusts the primary conclusion to reflect this. In that regard this is an improvement. It would still appear that the study limitations meaningfully limit study significance. That said, the issue remains an important one and the approach unique. 

Reviewer 2 Report

No more comments